# A View on the Chemical and Biological Attributes of Five Edible Fruits after Finishing Their Shelf Life: Studies on Caco-2 Cells

**DOI:** 10.3390/ijms25094848

**Published:** 2024-04-29

**Authors:** Lucia Camelia Pirvu, Nicoleta Rusu, Cristina Bazdoaca, Elena Androne, Georgeta Neagu, Adrian Albulescu

**Affiliations:** 1Department of Pharmaceutical Biotechnologies, National Institute of Chemical Pharmaceutical Research and Development, 112 Vitan Av., 031299 Bucharest, Romania; 2Department of Chemical Analysis and Drug Control, National Institute of Chemical Pharmaceutical Research and Development, 112 Vitan Av., 031299 Bucharest, Romania; nrusu@ncpri.ro (N.R.); cristinabazdoaca14@gmail.com (C.B.); androneelena51@gmail.com (E.A.); 3Department of Pharmacology, National Institute of Chemical Pharmaceutical Research and Development, 112 Vitan Av., 031299 Bucharest, Romania; georgetaneagu2008@gmail.com; 4Stefan S. Nicolau Institute of Virology, 285 Mihai Bravu Av., 030304 Bucharest, Romania

**Keywords:** perishable fruits, ethanolic extracts, chemical and biological changes, polyphenols dynamic, antioxidant activity, minerals partition, cytotoxicity and anti-proliferative, human tumor colon cell line Caco-2

## Abstract

We studied five common perishable fruits in terms of their polyphenols dynamic, minerals distribution, scavenger activity and the effects of 50% ethanolic extracts on the viability of Caco-2 cells in vitro, over a period of time between T = 0 and T = 5/7 days, typically the end of their shelf life. Altogether, there were few changes found, consisting of either an increase or a decrease in their chemical and biological attributes. A slow decrease was found in the antioxidant activity in apricot (−11%), plum (−6%) and strawberry (−4%) extracts, while cherry and green seedless table grape extracts gained 7% and 2% antioxidant potency, respectively; IC_50_ values ranged from 1.67 to 5.93 μg GAE/μL test extract. The cytotoxicity MTS assay at 24 h revealed the ability of all 50% ethanol fruit extracts to inhibit the Caco-2 cell viability; the inhibitory effects ranged from 49% to 83% and were measured at 28 µg GAE for strawberry extracts/EES, from 22 µg to 45 µg GAE for cherry extracts/EEC, from 7.58 to 15.16 µg GAE for apricot extracts/EEA, from 12.50 to 25.70 µg GAE for plum extracts/EEP and from 21.51 to 28.68 µg GAE for green table grape extracts/EEG. The MTS anti-proliferative assay (72 h) also revealed a stimulatory potency upon the Caco-2 viability, from 34% (EEA, EEG) and 48% (EEC) to 350% (EES) and 690% (EEP); therefore fruit juices can influence intestinal tumorigenesis in humans.

## 1. Introduction

Current statistics [1,2,3,4,5] show that the vegetal waste in the agri-food system costs EUR 12.4 billion per year in the United Kingdom alone; the Food and Agriculture Organization (FAO) in Europe, as well as the United States Department of Agriculture (USDA), report that 40 to 45% of the annual production of fruits and vegetables ends up as waste. Of those, it is estimated that 45% are losses at the agricultural stage, 30% at the post-harvest processing and distribution stage and 25% at the final consumer. Altogether, the data indicate that the vegetal waste from the agri-food system costs about USD 990 billion annually in the global economy [6]. Much more, the agri-food waste led to the loss of a quarter of the entire water used in the agriculture worldwide [7,8,9]. From the environmental point of view, the perishability of the agri-food products also results in substantial collateral assets’ losses, for example, large fertile areas of land and the energy for agricultural work performed on these cultivated areas. The agri-food losses also mean important emissions of greenhouse gases which in turn induce reductions in the plant variability and natural ecosystems biodiversity; at least 170 million tons of CO_2_ equivalent (3% of the total annual emissions) appear as the result of the life cycle of the agri-food products that end up as waste at the European level alone [7,8,9]).

As a result, current research is directed towards the identification of new green extraction technologies, and the recovery of the most valuable secondary metabolites in the vegetable wastes; due to their valuable content of secondary metabolites, grapes and olives appear to be on the top of the interest list [10,11,12,13,14,15,16]. Another focus point in the current research consists of understanding the relationships between the matrix and the secondary metabolites embedded in it, by both improving the extraction process and obtaining products with increased bioaccessibility and bioavailability in humans [17,18,19,20,21]. The need for using metabolomics for waste authentication must also be underlined; this is in the context in which the most frequent cases of food falsification occur the in top vegetable-derived products (e.g., olive oil). Therefore, the waste resulting after their technological processing is often contaminated with some cheaper substitutes [22,23]. The identification of a sufficient number of compounds in the vegetal waste to prove the lack of contaminants is all the more important as the extraction compounds are in the category of phytomedicines (e.g., food supplements, dietary supplements, drugs).

The result is that the fruits and vegetables which have exceeded their shelf life are the safest vegetal wastes, since they have not suffered any contamination through technological processing; and it would be highly useful to study the magnitude of the chemical modifications in these fruits and vegetables, and their influence on the main health attributes, in order to estimate the potential negative effects upon humans.

For these reasons, the present study has been focused on five edible fruits that are highly perishable but highly popular among consumers in Western countries, and at the same time they have high nutritional [24] and biological properties; these are strawberry, sweet cherry, apricot, plum and green seedless table grapes. According to statistics [25,26,27,28,29,30], the total consumption of these fruits in the EU is estimated at around 5.1 million tons of fresh fruit annually (1.2 tons strawberries, 0.7 tons sweet cherries, 0.5 tons apricots, 1 ton plums and 1.7 tons table grapes, respectively); applying the percentage of the vegetable waste worldwide (45%), this results in a loss of up to 2.3 million tons of fruit annually, which represents a lost opportunity to obtain pharmacological active ingredients, functional foods and food supplements too.

As is well-known, fruits are the most abundant source of active compounds in the series’ micronutrients and secondary metabolites; among the secondary metabolites, the anthocyanidins subclass is the common chemical denominator of most of the edible fruits and hence their common antioxidant properties and similar human health benefits. Apart from the anthocyanidins subclass, the edible fruits contain a plethora of other active phenolics (Table 1) able to offer particular, targeted biological effects in humans.

Altogether, the present study aimed to analyze the changes over the shelf-life period of five edible fruits of high interest in relation to their main biological attributes in humans: antioxidant activity and their impact on the cells with which they come into direct and prolonged contact in humans, namely the intestinal cells. Each fruit series was analyzed at the beginning of the experiment (t = 0) and after 5 days (strawberries and cherries) and 7 days (apricots, plums and grapes) of storage in conditions similar to those for the final consumer (in a refrigerator at 4 °C).

Briefly, the fresh fruits were divided into triplicate batches weighting around 100 g; each test batch was processed to obtain two fruit-derived products: the total acetone extractable material was further processed as 50% ethanolic extract (EE products) as the equivalent of the fruit juice (EES/strawberries, EEC/cherries, EEA/apricots, EEP/plums, EEG/green grapes), and the outcome of the acetone extractable material named the acetone powder (AP products) as the equivalent of the fruit waste (APS/strawberries, APC/cherries, APA/apricots, APP/plums, APG/green grapes); EE and AP series were studied at t = 0 and t = 5/7, respectively. The analytical studies were designed on the dynamics of polyphenols and minerals in the EE and AP; the pharmacological studies were designed on the EE products (the equivalent of fruit juice) and they consisted of cytotoxicity and antiproliferative MTS screening assays on the human colon tumor cell line Caco-2. The choice for these particular cells is motivated by the fact that the fruits come into direct and long-term contact with the intestinal cells upon ingestion; therefore, intestinal cells play a significant role in processing the biological active compounds found in fruits and more generally in all vegetal food.

## 2. Results

### 2.1. Chemical Analytical Results

The analytical studies were focused on the two main products obtained from the five fruits in the study: the acetone extracts prepared as 50% ethanolic extracts (EES, EEC, EEA, EEP, EEG) as the equivalent of the fruit juice, and the corresponding acetone powders (APS, APC, APA, APP, APG) as the equivalent of the fruit waste. As is well-known, fruit juice encompasses the major part of the micronutrients and secondary metabolites in fruits, while the outcome fruit waste contains the macronutrients and only traces of phenolics and micronutrients; therefore, the vegetal waste focuses the compounds with nutritional and microbiome value. The main goal of the analytical study was to analyze the dynamics of the polyphenols compounds in the five fruits extracts (EE products), before and after 5–7 days of shelf life in the refrigerator of the habitual consumer, for further connection with the results on their performance in hydroxyl radical scavenger activity and the effects upon human intestinal cells in vitro. The distribution of the minerals and microelements between the fruit juice and the fruit waste has also been analyzed, for better knowledge of the fruits’ waste losses too.

Table 2, Table 3 and Table 4 present the chemical quantitative aspects (total phenolics and total minerals and microelements contents) of the two type of products obtained from the five fruits series’ EE products and AP products, at t = 0 and t = 5/7 days, respectively.

The analysis of the five fruits’ phenolics’ dynamics along the 5–7 days of refrigerated storage indicated that the extracts from cherries/EEC and green seedless table grapes/EEG lost small quantities of total phenolics content along the shelf life (precisely a 1.72% and 7.74% decrease in comparison with the total phenolics at t = 0), while the strawberry/EES, plum/EEP and apricot/EEA fruit extracts revealed an augmentation of the total phenolics content at the end of the experiment (precisely a 4.4 0%, 6.15% and 11.04% increase in comparison with the total phenolics at t = 0).

These quantitative variations could be explained by oxidation–precipitation processes along the storage and extraction processes; thus, cherries and green table grapes suggested a decrease in polyphenols content after 5–7 days of storage in the refrigerator, plums indicated a gain in polyphenols percentage along the shelf life, while strawberries and apricots indicated variations which were compensated by the dehydration percentage. However, these results have to be seen in the context of the main biological activity of the plant phenolics, the reactive oxygen species scavenger efficacy, since the oxidation, isomerization and hydrolysis processes occurred along the extraction processes leading to compounds with a different (smaller or bigger) antioxidant efficacy.

Table 3 presents the content of minerals and microelements in the acetone powder series (APS, APC, APE, APP, APG products) to achieve an image of their content in the vegetal waste, after acquiring the fruit juice. The measurements are at t = 0 only, since there is no reason for the loss of the minerals and microelements during the shelf life.

Table 4 compares the data from the literature [50] with the data from the present study (Table 3), with the purpose of concluding the distribution of the main elements in the two main parts of the five fruits in the study: the fruit extract and the acetone powder/waste.

The computed results suggest that the potassium element mostly passes into the acetone extract and therefore into the 50% ethanolic extract; therefore, the fruit juice likely concentrates from 70% to 83% of the potassium in the fresh fruits. The calcium element appears as more dependent on the fruit matrix; therefore, the strawberry waste indicated the most abundant quantity of the calcium in the acetone powder (about 73%), the plum waste retained up to 50% of the calcium, cherry waste up to 45% and the apricot and grape wastes up to 38%. The magnesium element preferred the matrix part from cherries and strawberries (55% and 46% in waste), and the juice part from plums, apricots and grapes (71%, 67% and 63% in extracts). The iron element indicates a preference for the strawberry waste which appeared to retain up to 82% of the total iron in the fresh fruit. Altogether, the main elements in the five fruits series, K, Ca, Mg and Fe, mostly passed into the fruit juice; the acetone powders, meaning the equivalent of the fruit waste, revealed 22 to 31% of these elements. Yet, if potassium is abstracted from the calculation, the fruit waste retains half of the Ca, Mg and Fe minerals in the fresh fruits, so it can be concluded that the fruit waste is an important source of minerals and trace elements, very affordable and safe for the human use.

Furthermore, Table 3 shows the presence of the heavy metals in the five edible fruits in the study (Pb and As). The literature data currently draw attention to the fact that fruits and vegetables are frequently contaminated with heavy metals. Lead (Pb), arsenic (As), mercury (Hg) and cadmium (Cd) are considered the most dangerous heavy metals for humans, and all the more so in children [51,52,53,54]. For example, lead (Pb) ingestion is associated with weakened skeletal, immune and endocrine deficits and lower intellectual capacity in children, and hypertension and renal and heart dysfunctions in adults [55,56]; the trivalent inorganic As is considered the most hazardous form in nature, with a multitude of harmful effects in humans, including carcinogenic effects [57]. The document named the “Codex general standard for contaminants and toxins in food and feed (CODEX STAN 193–1995)” [58] indicates the safety levels of the main contaminants and toxins in food and feed; in the case of the two heavy metals, these are admitted contents up to 0.1 mg/kg for As and up to 1 mg/kg for Pb.

In the present study, the levels of the two heavy metals (Pb, As) were in the interval from 0.02 to 0.06 mg/kg, FW; cherries contained the highest content of the two heavy metals (0.01104 mg%), followed by apricots (0.00836 mg%), strawberries (0.00806 mg%), green seedless grapes (0.00588 mg%) and plums (0.00561 mg%), FW.

These data have to be seen in the context in which polyphenols and minerals are embedded in the fruit matrix; their (bio)accessibility is therefore dependent on a multitude of physical, chemical and enzymatical processes along the digestion process. Much more, studies have revealed a cooperation between the matrix and the secondary metabolites in plant products. For example, it was proved [16] that lipids and flavonoids may enhance the carotenoids’ bioaccesibility, while minerals may decrease this process; the dietary fibers and proteins are also likely to affect carotenoids bioaccessibility, as well as the copresence of the biliary salts and lipases. Other studies aimed to assess the bioavailability of several polyphenols compounds, particularly anthocyanins and hydroxycinnamic acids, and carotenoids compounds from peaches, plums, prunes, walnuts and tomatoes showed a progressive release of the total phytochemicals along the digestion process [17,18]. Thus, anthocyanins were proved as being the most affected compounds by oral digestion; anthocyanins also presented the best bioaccessibility in humans. Hydroxycinnamic-acid derivatives were shown to have a higher stability during the digestion process. Studies have also concluded that when there is a much higher content of flavonoids in the matrix, the total phenolic content is higher after the digestion simulation process [19]; for example, it was proved that gelatin provides the best release of phenolic acids, flavonoids and anthocyanins in the environment, at the same time resulting in the highest antioxidant activity [20]. Furthermore, in vitro gastrointestinal digestion assays on several dried fruits indicated that Na, K, Mg, Fe, Mn and Cu presented moderate bioaccessibility, while Zn did not present bioaccessibility in the simulation study [21].

In close relation to the use of metabolomics for vegetal-derived products’ chemical characterization and waste authentication [22,23], studies underline that “In quantitative analysis of environmental samples using high-performance liquid chromatography-electrospray ionization mass spectrometry (HPLC-ESI-MS) one of the major problems is the suppression or, less frequently, the enhancement of the analyte signals in the presence of matrix components” [59]. Accordingly, three calibration approaches to compensate the matrix effects (ME) are recommended: external sample calibration, internal sample calibration with one standard and external sample calibration corresponding to matrix-matched calibration. The results indicated that the external sample calibration can be suitable to compensate the matrix effects from moderately loaded samples with more uniform matrices (the resulting deviations were below 25% as compared to standard addition). Applied to the field of natural derived products, food-science and natural medicines too, the need for a combination of mass spectrometry techniques (MS) with high-performance liquid chromatography (HPLC) to achieve the chemical characterization of the vegetal products further the MS limitations induced by the matrix effects (MEs) [60]; it was concluded that the analyte co-elutes together with other molecules in the vegetal samples to alter the quantitative analytical results, which is detrimental during the method validation by negatively affecting the reproducibility, linearity, selectivity, accuracy and sensitivity of the method. Accordingly, the tutorial review [60] recommends that when the sensitivity is crucial, the analysis must try to minimize ME by adjusting MS parameters, chromatographic conditions or optimizing clean-up; otherwise, satisfactory results by ME compensation can be obtained using three calibration approaches, depending on the availability of the blank matrix; thus, when blank matrices are available, the methods of isotope-labeled internal standards and matrix-matched calibration standards can be used; when blank matrices are not available, the calibration method using the isotope-labeled internal standards, background subtraction or surrogate matrices is recommended. Furthermore, the adjusting of MS parameters, chromatographic conditions and clean-up steps were all proved as necessary, while minimizing MEs, and not just compensating for them, was the authors’ recommendation in both study cases [60]. The measures recommended are (1) sample dilution and smaller injection volumes resulting in reduced matrix suppression, and the possibility of using standards in neat solvent as a calibration method [61]; (2) keeping the preparation and/or chromatographic analytical procedure unmodified [60]; (3) using appropriate chromatographic conditions to improve the separation of the analyte from interfering substances and avoiding its co-elution with the matrix [62]; and (4) clean-up optimization to reduce the amount of the matrix components introduced into the analytical system [63,64,65]. Finally, the recommendations for the reduction in the matrix components are the removal of proteins using common methods for protein precipitation such as salting out and precipitation with organic solvents (e.g., acetonitrile); the removal of phospholipids using common defatting procedures (e.g., hexane extraction or freezing lipid precipitation) and the removal of sugars, the most common interference species mostly in food matrices, using a solid adsorbent such as primary secondary amines/PSA [60].

Related to the present study, by using acetone solvent at increasing concentration levels in the extraction stage, followed by concentration of the total acetone extract, residuum resumption in 50% ethanol and filtration on the double paper filter, in fact most of the proteins, soluble polysaccharides and lipids in the fruits matrix were removed, but not the small sugars which are soluble in both solvents used. This means their potential interference in the analytical assessments and in vitro MTS screening studies too.

### 2.2. Antioxidant Activity Results

Studies were carried out by the chemiluminescence (CL) method using Luminol—H_2_O_2_ system in basic medium (pH = 8.6), a very sensitive and reproducible method through which the authors have studied a large number of plant extracts [66,67].

Tests were conducted on the five series of 50% ethanolic extracts (EES, EEC, EEA, EEP, EEG) at the beginning (t = 0) and at the end of the experiment (t = 5/7), in comparison with the negative control sample in the study (the 50% ethanol solvent, *w/v*). The five extracts (EES, EEC, EEA, EEP and EEG) at t = 0 and t = 5/7 were each prepared as 8 dilution series: ×1, ×5, ×10, ×20, ×40, ×80, ×160, ×320, in triplicate. Each 50 μL test extract and dilution point in the series was mixed with 200 μL of 10^−5^ M Luminol and 700 μL 0.2 M Tris-HCl pH 8.6; the chemiluminescence reaction was initiated by adding 50 μL 10^−3^ M H_2_O_2_ (fresh solution) in the medium. The CL reaction (arbitrary units/a.u.) was measured in a chemiluminometer at intervals of 5 s along 60 s total time. The results are compared with the negative control series. Thus, the antioxidant activity (AA%) of each test sample can be analyzed at exactly 5 s after the CL reaction initiation in comparison with the activity at 5 s of the negative control series (see formula below); the IC_50_ value of each test sample, found by achieving the inflection point at which the maximum antioxidant activity at the minimum concentration of the test product, is also available.
(1)$$\mathrm{AA}\%=\frac{a.u.\;\mathrm{control}\;\mathrm{sample}\;-\;a.u.\;\mathrm{test}\;\mathrm{sample}}{a.u.\;\mathrm{control}\;\mathrm{sample}}\times 100$$

Appendix A presents the dynamic of the hydroxyl radical scavenger activity along the 60 s of the CL reaction, for each one of the five series of 50% ethanolic extracts in the study (EES, EEC, EEA, EEP, EEG), the mean values for each dilution point in the series at the initiation time (t = 0) and at the end of experiment (t = 5/7), in comparison with the negative control series (*n* = 3, ± SD).

Table 5 presents the AA% and IC_50_ values of the five series of 50% ethanolic extracts from the five fruits’ series, at t = 0 and t = 5/7.

Overall, the CL method indicated the high antioxidant and hydroxyl radical scavenger efficacy of the tested extracts; the IC_50_ values ranged from 1.67 to 5.93 μg gallic acid equivalents [GAE] per 1 μL test extract. The most augmented antioxidant efficacy has been calculated in the case of extracts from apricots, followed by plums, strawberries, green table grapes and cherries.

The comparison of the IC_50_ values at the beginning and at the end of experiment indicated the following changes in the reactive oxygen species (ROS) scavenger activity of the five fruits series tested: there were revealed decreases in ROS scavenger efficacy in the case of 50% ethanolic extracts from apricots (−11%), plums (−6%) and strawberries (−4%); while in the case of 50% ethanolic extracts from cherries and green seedless table grapes, increases in ROS scavenger activity, with 7% and 2%, respectively, were computed.

Obtaining an antioxidant activity of 99% magnitude indicates in fact the exceptional efficiency of the (ethanolic) extracts from the edible fruits against the hydroxyl radicals in the environment; from our practice obtained on over 30 plant species, the extracts from the medicinal plants with similar IC_50_ values (1–6 μg GAE/1 μL ethanolic extract) usually result in antioxidant activities from 80% to 90%.

### 2.3. In Vitro Pharmacological Activity Results

In vitro pharmacological studies were carried out on the five series of 50% ethanolic extract (EES/strawberries, EEC/cherries, EEA/apricots, EEP/plums, EEG/green seedless table grapes), for the variants at the beginning of the experiment (t = 0) and at the end of experiment (t = 5/7). As presented, due to their relevance and responsiveness to plant-derived products, the tests were conducted on the human colon tumor Caco2 cell line (ATCC, HTB-37) following the Promega MTS [3-(4,5-dimethyl-thiazol-2-yl)-5-(3-carboxy- methoxyphenyl)-2-(4-sulfophenyl)-2H–tetrazolium] protocol, cytotoxicity and anti-proliferative screening assays, respectively [68]. The in vitro assays were designed in the range of the biological activity of the plant compounds, covering the concentrations from 1 to 55 µg gallic acid equivalents [GAE] per sample. This way, each fruit extract in the study (EE products), the variants at t = 0 and t = 5/7, was prepared as a triplicate five-dilution series, using 20% heat-inactivated fetal bovine serum (the Caco-2 cell growth medium); in parallel, the positive control series (using 50% ethanol) and the negative control series (using 20% heat-inactivated fetal bovine serum) were prepared in identical conditions. The resulting test/control samples series were studied on Caco-2 cells (ATCC, HTB-37), by cytotoxicity and anti-proliferative MTS assays, respectively; the difference between the two types of assays consists of the fact that the cytotoxicity assay uses the “semi-confluent” cell culture type (up to 70% cell proliferation) while the anti-proliferative assay uses the “sub-confluent” cell culture type (up to 30% cell proliferation). After 20 h (cytotoxicity assay) and 44 and 68 h (anti-proliferative assay) of cell exposure to the test/control series samples, the culture medium was removed and the cells were incubated with MTS solution for another 2 h; at the end, the viability of the adherent cells was measured by reading the solution’s absorbance at 492 nm. The O.D. values at 492 nm for each dilution point test and positive and negative control series were computed for their statistical significance (“*t*” test); the mean values (*n* = 3) were computed as O.D. 492 nm and cell viability percentage (%) along the dilution series expressed as μL test sample and μg GAE per mL sample.

Statistic notations: in the present study, the notation * means results without statistical significance and *p* > 0.05; the notation ** means results with statistical significance and 0.05 < *p* < 0.01; the notation *** means results with statistical significance and *p* < 0.01.

#### 2.3.1. Cytotoxicity MTS Assay Results on the Five Series of Fruits Extracts, at t = 0 and t = 5/7

Figure 1A–E presents the in vitro cytotoxicity activity results of the five series of 50% ethanolic extracts (EES, EEC, EEA, EEP, EEG), the variants at t = 0 and t = 5/7, tested on Caco-2 cells after 24 h of cell exposure.

By comparison with the negative control series (the gray line), the MTS cytotoxicity assay at 24 h overall indicates the ability of all tested fruits’ extracts to inhibit the viability the human colon tumor cell line Caco-2 in vitro. The differences between the pairs in the study, the extract at t = 0 and t = 5/7, respectively, were significant, but not in the point or in the interval of their maximum inhibitory effectiveness; except for the strawberry extracts, the differences registered were less than 5%. By comparison with the positive control series (the blue line), the extracts from strawberries and green seedless table grapes were emphasized as very effective protectors against the negative effects of the ethanol solvent upon the human colon cells growth in vitro.

By corroborating the polyphenols’ concentration (Table 2) in the five series of test extracts (EE at t = 0 and t = 7), the maximum inhibitory effects were estimated at 28 µg GAE/mL sample in the case of strawberry extracts/EES (77% vs. 61%), at 22–45 µg GAE/mL sample in the case of cherry extracts/EEC (79% vs. 74%), at 7.58–15.16 µg GAE/mL sample in the case of apricot extracts/EEA (77% vs. 78%), at 12.50–25.70 µg GAE/mL sample in the case of plum extracts/EEA (52% vs. 49%) and at 21.51–28.68 µg GAE/mL sample in the case of green seedless table grape extracts/EEG (79% vs. 83%).

Similarly, with the exception of the green seedless table grapes/EEG extracts, for the 50% ethanol extracts from cherries/EEC, apricots/EEA and even more augmented in the case of plums/EEP and strawberries/EES, an inflection point along the dilution series was noticed. Obtaining an inflection point in the concentration range that can usually be reached through the consumption of fruit juice by humans resulted in the anti-proliferative MTS assay being carried out over a longer exposure of Caco-2 cells to the five fruits’ extracts, at 48 and 72 h, respectively.

#### 2.3.2. Anti-Proliferative Assay on the Five Series of 50% Ethanolic Extracts, at t = 0 and t = 5/7

Figure 2A–J presents in vitro anti-proliferative MTS assay activity results of the five series of 50% ethanolic extracts (EES, EEC, EEA, EEP, EEG), the variants at t = 0 and t = 5/7, tested on Caco-2 cells and measured at 48 h and 72 h, respectively.

The anti-proliferative MTS assay at 48 and 72 h of cell exposure to the test extracts overall revealed the stimulatory potency of the five fruits extracts in the study upon the viability of the human carcinoma tumor cell line Caco-2 in vitro; the stimulatory effects started at the inflection point noticed in the cytotoxicity MTS assay; as expected, the most augmented stimulatory effects were revealed in the case of the extracts from strawberries (A,B) and plums (G,H), but the extracts from cherries (C,D), apricots (E,F) and green table grapes (I,J) also showed significant stimulating effects at the maximum GAE concentration in the series samples.

The computation of the O.D. at 72 h in comparison with both the negative control series (blue line) and the positive control series (red line), which became identical at 72 h, together revealed the magnitude of the stimulatory effects upon the Caco-2 cells; they were up to 690% at 51 µg GAE plum extracts, up to 350% at 55 µg GAE strawberry extracts, up to 48% at 45 µg GAE cherry extracts and up to 34% at 15 and 29 µg GAE apricot and green table grape extracts, respectively (mean values, at t = 0 and t = 5/7).

Altogether, the anti-proliferative MTS assay suggested that the ethanolic extracts, and therefore the juice part from the edible fruits, particularly the strawberries and plum juices, could have an augmented stimulatory potency upon (tumor) colon cells in humans.

## 3. Discussion

As is well-known, dietary flavonoids are represented by six major subclasses in nature, namely flavonols, flavones, flavanones, flavan-3-ols, anthocyanins and isoflavones. They are mostly derived from fruits, vegetables, cocoa and tea consumption, and the total polyphenols intake in humans is estimated at 1 g daily.

Also, apart from their notorious antioxidant anti-inflammatory activities, the dietary flavonoids are renowned in promoting metabolic and cardiovascular health, in promoting cognitive and vascular endothelial functions, as well as in sustaining the glycemic response in type 2 diabetes mellitus, and they are proved to decrease the risk of breast cancer in postmenopausal women too [69]. Above these, the literature data in recent years [70,71,72,73,74,75,76,77,78,79,80,81] point out the existence of a microbiota–flavonoid bidirectional interaction meaning that the gut microbiota influence the fate of the flavonoids regarding their metabolic mechanism, biotransformation, bioavailability and bioactivity in humans, but also flavonoids influence the human microbiome’s homeostasis as well. Furthermore, flavonoids can regulate the human microbiota by promoting the growth of beneficial bacteria (prebiotic effect) and by inhibiting opportunistic and pathogenic bacteria [72,73]. It has been established that the microbial metabolism of the dietary flavonoids follows a general model by which the bacterial enzymes carry out the -*O*-deglycosylation of the flavonoids glycosides to release their active aglycones, and at the same time, the gut bacteria perform many other transformations, including oxidation and demethylation reactions, which result in smaller molecules in the series of phenolic acids [70,82,83,84]. According to the data reported, the main flavonoid-converting bacteria in the human microbiome are in the *Actinobacteria*, *Bacteroidetes* and *Firmicutes* genera, plus several strains of *Proteobacteria*. *Lachnospiraceae* stand out as the main flavonoid-*O/C*-deglycosylation strains in humans [70,71,72,73]. At the same time, studies on animal models [74,75,76,77] indicated that quercetin [74], quercetin and resveratrol [75] and myricetin [76] have the ability to reshape the intestinal flora disorders in high-fat diet (HFD)-induced non-alcoholic fatty liver disease (NAFLD) mice models exactly by decreasing the ratio of *Firmicutes*/*Bacteroidetes* and by increasing the level of *Akkermansia muciniphila* [76,77,78]. In the specific case of the myricetin compound, the protective activity against HFD dysbiosis was ascribed to the ability to modulate the level of the fecal butyric acid, and by offering the protection of the gut barrier function: two main mechanisms behind the regulation of the gut microbiome [76]. Furthermore, some clinical studies on 66 healthy men [79] seeking to assess the microbiome effects of berries indicated that the aronia fruits (the richest source of dietary anthocyanidins) resulted in a high microbial diversity; other studies on blueberry extracts on mixed cultures of human fecal bacteria indicated a significant increase in the number of *Lactic acid bacteria* and *Bifidobacteria* [80].

Altogether, the scientific data today converge towards the following mechanism of dietary flavonoids behind regulating the gut microbiota in humans: (a) the stimulation of the production of the short-chain fatty acids (SCFAs) from flavonoid–*O*-glucosides in undigested food (noticed in the case of puerarin, baicalin, hesperetin, genistein, rutin, quercetin and epigallocatechin-3-gallate) which further results in “upregulating the production of IL-22 by promoting the expression of aromatic hydrocarbon receptors and hypoxia-inducible factor 1, thus also protecting the intestine from the influence of inflammation” [85]; (b) the stimulation of the microbiotic gut–brain axis through the neuroendocrine–immune pathway, observed in obesity experiments [86]; (c) the stimulation of the number of the mucin-secreting goblet cells which have the ability to restore the colonic epithelial mucus layer, thus providing a suitable ecological niche for mucosa-associated symbiotic bacteria [87,88,89]; (d) by immune and inflammatory response stimulation [90]; (e) by maintaining the intestinal immune tolerance and gut health noticed in the case of quercetin [91]; and (f) by gut-associated lymphoid tissue stimulation, meaning a prebiotic-like activity, in the specific case of hesperidin [92].

On the other side, the edible fruits contain an enormous number of secondary metabolites, especially polyphenols compounds with well-established therapeutic properties. The common and often dominant presence of the anthocyanins and anthocyanidins derivatives makes the antioxidant activity the main biological and pharmacological attribute of the edible fruits; anthocyanidins’ combination with other flavonoids subclasses results in other multiple pharmacological effects, and therefore in holistic human health benefits. Finally, the polyphenols’ combination with some specific micronutrients, such as minerals, amino-acids, phytosterols, carotenoids and other polar and non-polar vitamins found in a particular fruit, lead to a virtuous chemical combination able to induce both particular benefits upon a specific tissue and organ in humans. Altogether, three main directions of valorization of the secondary metabolites in the fruits wastes are achievable: obtaining holistic antioxidants, the direction of targeted fruit-derived products for a certain tissue or organ on which they have the maximum pharmacological effect and the direction of the antitumor products.

Proving these, apart from the holistic benefits of the products from strawberries (e.g., fresh juice, beverages, jellies and jams, lyophilized powders and other concentrated products) [93,94,95,96,97,98,99,100,101,102,103,104], the literature data also indicate their particular benefits [97,98] upon blood cells (e.g., platelets, erythrocytes, mononuclear cells) and the function of the mitochondria; their allergic reactions are carried out exactly through the stimulation of the mast cells, basophils and eosinophils in the circulatory system, by the allergenic protein named Fra a 1 [98]. Strawberry-derived products also are some of the most effective ROS scavengers, being active toward many types of chemically generated radicals in a medium, e.g., peroxyl (ROO*), superoxide (O^2−^), hydrogen peroxide (H_2_O_2_), hydroxyl (*OH) and singlet oxygen (1O2) [99,100,101,102]. Regarding the antitumor activity, studies on a polyphenol-rich dry extract from strawberries tested on human dermal fibroblasts (HDF), murine macrophages (RAW264.7) and human liver carcinoma (HepG2), three cell lines representative in relation to the human body response to cytotoxicity, inflammation and nutrition [103], indicated stimulatory but also inhibitory activity upon the viability of the tested cells; in the case of HDF cells, after 24 h of exposure an increase was noticed in the cell viability up to the level of 10,000 µg dry extract/sample (computed at 157% stimulatory activity by comparison with the negative control sample), in a dose-dependent manner; after 48 and 72 h of cell exposure, cytotoxic effects in the interval from 2500 to 5000 µg dry extract/mL sample have been noticed. The RAW line, after 24 h of cell exposure, indicated an increase in the cell viability up to 1000 µg dry extract/sample (computed at 24% stimulatory activity by comparison with the negative control sample), while at 48 and 72 h of exposure, the viability of the cell line decreased in a dose-dependent manner (−25%, in comparison with the negative control sample); in the interval from 2500 to 7500 µg dry extract/sample, there were noticeable cytotoxic effects. The HepG2 line, after 24 h of exposure, indicated no significant effects up to 7500 µg dry extract/sample, while during longer exposure (48 and 72 h, respectively) cytotoxic effects in the interval between 250 and 1000 µg dry extract/sample were noticed. Other studies [104] on human breast (MCF-7) and human colon (HT29) cancer cell lines also indicated that starting with the concentration level of 500 µg dry extract/sample caused a decrease in the cell viability for both tumor lines; the inhibitory effects for the highest concentrations in the study ranged from 41% to 63% for the HT29 line and from 26% to 56% for the MCF-7 line. The MTS cytotoxicity assay on Caco-2 cells in the present study has revealed that after 24 h of cell exposure to 50% ethanolic extracts from fresh strawberries (EES at t = 0 and t = 5), both extracts inhibited the human colon carcinoma cells’ viability along the interval from 6 to 55 µg GAE/mL test sample. Also, the MTS study has shown an inflection point around 28 µg GAE/mL test sample; specifically, the Caco-2 cell viability decreased from 6 to 28 µg GAE/mL, after which it started to increase, but under the negative control line in the study; the maximum inhibitory effect noticed at 28 µg GAE/mL sample achieved the magnitude of 77% at t = 0 and of 61% at t = 5, respectively. The MTS anti-proliferative assay at 48 and 72 h indicated that starting with the inflection point noticed in the cytotoxicity study (28 µg GAE/mL sample), EES at t = 0 and t = 5 both induced stimulatory effects upon the Caco-2 line viability; the stimulatory effects were similar and reached 350% magnitude at the maximum concentration in the study (55 µg GAE/mL sample, mean value, *n* = 6), by comparison with both the negative control and positive control series in the study.

The products from cherries are also attributed with holistic human health benefits [105,106] and particular benefits upon the function of the brain [107,108,109,110,111,112,113,114,115,116,117,118,119,120] and the muscular system [121,122,123,124,125,126,127,128,129]. The benefits upon brain function are the results of the virtuous combination from anthocyanidins, tryptophan, serotonin and melatonin [107,108,109], and they consist of improving the quantity and the quality of the sleep, in improving the mood and in improving the cognitive function and counteracting the neuronal autophagy and amyloid beta production in Alzheimer’s disease [119,120]. The muscular benefits result from the inhibition of the oxidative stress and inflammatory response at the level of the muscle, particularly by decreasing the interleukin (IL6) production, at the same time as from the anthocyanidins’ benefits on the vascular system and the management of the creatinine, total proteins and cortisol levels in serum, which together mitigate damage in the overstretched muscle; these effects have been fully demonstrated in the particular case of sportsmen who suffer from these muscle overloads during sports competitions [121,122,123,124,125,126,127,128,129]. Concerning the antitumor potency, the MTS assays [130] performed on extracts from cherries in quantities of 50, 100, 200, 400 and 800 µg per sample applied on gastric adenocarcinoma (AGS), normal human dermal fibroblasts (NHDF) and neuroblastoma (SH-SY5Y) cell lines together indicated the AGS cells as the most responsive (IC_50_ = 130.39 ± 1.73, µg/mL); MTT assays on AGS cells revealed a dose-dependent response and also major DNA damage in the case of the most colored cherry extracts. At the highest concentration in the study (800 µg/mL), the most colored fractions proved the ability to induce necrosis in the AGS cells, consisting of mitochondrial and cellular swelling and plasma membrane disruption too; at the lowest concentration in the study (200 µg/mL), other morphological changes were observed associated with the apoptosis process, for example, condensed chromatin and fragmented nuclei. It must be noted that identical apoptotic effects were observed in the case of AGS exposure to quercetin (160 µM) [131], cyanidin-3-*O*-rutinoside (100 µM) and catechin (50 µM); the effects were similar in the case of the HepG2 and highly invasive breast cancer cell line (MDA-MB-231) [132]. Other studies on HepG2 cells [133] have revealed that after 72 h of cell exposure to sweet cherry extracts in concentrations of 25, 50 and 100 µg per sample, there resulted a decrease in the mitochondrial activity, in a dose-dependent manner (IC_50_ = 27.24 ± 0.72, µg/mL); the most significant effects were observed in the case of the most colored cherry extracts. An anthocyanin-rich fraction isolated from sweet cherries (800 µg/sample) also induced necrosis in the human colon carcinoma cell line Caco-2 [134], while another two cherry extracts tested on colon carcinoma (HT-29) inhibited up to 93.91% (IC_50_ = 2.5, μg/mL) and up to 94.25% (IC_50_ = 2.1, μg/mL) of the HT-29 cell line growth in vitro [135]. The MTS cytotoxicity assays in the present study also revealed the ability of 50% ethanolic extracts from fresh cherries (EEC) to inhibit the viability of Caco-2 cells; the interval tested ranged from 5 to 45 µg GAE/mL sample, while the maximum inhibitory potency (79% and 74% at t = 0 and t = 5, respectively) was manifested in the interval from 22 to 45 µg GAE/mL sample. The MTS anti-proliferative assays at 48 and 72 h of exposure to EEC revealed that starting with the concentration level of 22 µg GAE/mL, the EEC at t = 0 and t = 5 both induced the stimulation of human colon tumor cell viability in vitro; by comparison with the negative and positive control series in the study, the effects on Caco-2 cells at 72 h and the maximum concentration level (44 µg GAE/mL, mean value, *n* = 6) were estimated at up to 48% stimulatory potency.

The apricots appear with holistic but also with targeted benefits upon the eye and stomach functions [44,45,136,137,138]. The eye benefits are based on the large palette of carotenoids derivatives present in apricot fruits [139,140]. As is well-known, the center of the eye retina, named the macula lutea, contains a multitude of carotenoid compounds (called macular xanthophylls) in the series of lutein, zeaxanthin and meso-zeaxanthin. Macula carotenoids absorb the light from the visible region (400–500 nm), thus offering retina and lens protection against the photochemical damage induced by light exposure. Carotenoids are also very effective scavenger compounds of the reactive oxygen species in the medium, this being especially important for eye health and function; their placement in the center of the retina is exactly for the purpose of counteracting the damaging light, oxidative stress, apoptosis, mitochondrial dysfunction and eye inflammation induced by external and internal aggressors, as well. Proving this, the main eye dysfunctions in humans (such as age-related macular degeneration, cataract, diabetic retinopathy, retinitis pigmentosa and retinopathy of prematurity) are related to a low carotenoids level in the retina, particularly of the lutein compound [141]. The stomach benefits are mainly achieved through the antimicrobial properties of the apricots, more precisely by the inhibition of the motility and colonization of *Helicobacter pylori* in the stomach wall [44,142]. The compound active against *H. pylori* motility and colonization is a syringic-acid derivative, named syringaresinol; proving this, a high intake of apricot fruits resulted in a significant reduction in *H. pylori* load and mononuclear and neutrophil infiltration in the gastric mucosa, at the same time preventing chronic atrophic gastritis by reducing the active mucosal inflammation in a clinic experiment on non-elderly individuals [142]. The antimicrobial properties were also proved in the case of oral cavity and periodontal infections with *Aggregatibacter actinomycetemcomitans* and *Porphyromonas gingivalis* [137]. Also, a combination of citric acid, chlorogenic acid, isoquercitrin asparagine and epicatechin isolated from the ethanolic extracts of fruits of *Prunus mume* was very effective against the strains *Listeria monocytogenes*, *Salmonella enterica*, *Bacillus cereus*, *Staphylococcus aureus* and *Escherichia coli* [143]; furthermore, studies on the separate compounds have revealed that citric acid had a higher-ranking activity against all five bacteria in the study, chlorogenic acid inhibited four of the five bacteria studied, epicatechin inhibited *Salmonella enterica* and *Listeria monocytogenes*, while asparagine and isoquercitrin inhibited *Listeria monocytogenes* only [143]. Studies regarding the antioxidant activity of the products from apricots indicated a high antioxidant potency [136,137,138,144,145]; the decrease in the reactive oxygen species scavenging efficacy after processing apricot fruits has also been established [146]. Concerning the antitumor activity, the highest interest of today is upon the extracts from the kernels of apricots since they contain a very active antitumor compound named amygdalin [44,45,147]. The fresh fruits, as well as the extracts from the apricots, both were proved to have antitumor potency [148]; an in vivo study [149] on rat groups exposed to radiotherapy and 7,12-dimethylbenz(a)anthracene/DMBA in order to induce tumors in liver showed that a daily supplementation of 20% apricot fruits counteracted the oxidative damage in treated rats; also, measurements showed a reduced level of alanine aminotransferase (ALT), aspartate aminotransferase (AST), 5′-nucleotidase (5′NT), malondialdehyde (MDA) and nitric oxide (NO), a reduced expression of B-cell lymphoma (Bcl-2) protein and of activator protein 1 (AP-1), of the cyclic AMP (cAMP) response element-binding protein (CREB) and of the cytokine nuclear factor kappa B (NF-κB) level in the treated rats; increases in the activity of the antitumor modulators Bax and caspase-3 appoptosis proteins and reduced glutathione (GSH) were also noticed, while the histopathological examinations of the liver samples revealed that mitosis, pericentral necrosis and pleomorphism caused by DMBA exposure in rats were mitigated after the apricot and/or radiotherapy administration [149]. Also, an ethanolic extract from apricots (4 mg/sample) proved the ability to inhibit the viability of the human gastric carcinoma/AGS (up to 58%), human colon cancer/MCF-7 (up to 72.8%), human lung carcinoma/A549 (up to 88.2%) and human cervical adenocarcinoma/HeLa (up to 89.4%) [150]. The present MTS cytotoxicity assay at 24 h indicated the augmented inhibitory effects of the 50% ethanolic extracts obtained from fresh apricots (EEA, at t = 0 and t = 7) on the viability of the human colon carcinoma cell line Caco-2; the interval tested ranged from 1.89 µg GAE to 15.16 µg GAE/mL sample, while the maximum inhibitory potency (computed at 77% and 78% at t = 0 and t = 7, respectively) was noticed in the interval from 7.58 to 15.16 µg GAE/mL sample. The MTS anti-proliferative assays on Caco-2 cells at 48 and 72 h indicated that starting with the concentration level of 7.58 µg GAE/mL sample, EEA at t = 0 and t = 7 started to stimulate the viability of the human colon tumor cells in vitro; at 72 h of cell exposure to the maximum concentration level in the study (15 µg GAE/mL sample, mean value, *n* = 6), the stimulatory potency achieved up to 34% magnitude, by comparison with both the negative and the positive control series in the study.

Apart from their holistic effects [151,152,153], plums have been proven to mitigate acute gouty arthritis and macrophage-induced inflammation in joints [154,155]; at the same time they can induce cognitive and bone mass improvement in humans [156,157,158,159,160]. As is well-known, the cause of acute gouty arthritis is monosodiumurate (MSU) crystals’ deposition in the joints which further induces the stimulation of the phagocytosis of macrophages [154]. Studies on a polyphenol-rich extract from plums (681 mg GAE/g test product) tested on an MSU-stimulated RAW264.7 macrophage cell line model in vitro indicated significant anti-inflammatory activity; the effects were achieved through superoxide dismutase (SOD) activation and by inhibiting tumor necrosis factor alpha (TNF-α) and interleukins (IL-1β, IL-18) pro-inflammatory cytokines, and the anti-inflammatory effects, SOD and malondyaldehide (MDA) activities, respectively; each one followed a dose–effect relationship and achieved the maximum potency at 60 µg plum fruits extract per mL sample [155]. Regarding the benefits on cognitive function in humans, studies on rat groups receiving a high-cholesterol diet plus 2% and 5% plum powder indicated a significant difference in the time taken to complete the Morris water maze task in the case of treated rats; the effects were significant in comparison with both control groups, the positive control group fed with a high-cholesterol diet and the negative control group too [156]. The comparison between the effects induced by the ingestion of two variants of plum powder (2%), the first one obtained from the fresh plum juice and the second from dried plums, indicated that only the plum juice powder induced the improvement in cognition, not the produce from prunes (the dried plums) [157]. Chlorogenic acid, the main phenolic acid in plums, was proven to be the basis of the anti-anxiolytic effects noticed in a light/dark exposure mice model too; according to the data obtained, chlorogenic acid protects the granulocytes against the oxidative stress and degranulation induced by a light/dark model in mice [158]. The consumption of plums was also effective against the oxidative stress induced by irradiation [159], while the bone health benefits are explained by the effective increasing of the calcium retention in bone cells (achieving up to 20% efficacy) [160]. The antioxidant efficacy [161] and antitumor potency [162] of plums were also studied. The reviewed data indicate the efficacy of the plum-derived products on more than 20 human tumor cell lines [162], e.g., on the human hepatoma cell line HepG2, human cervical carcinoma cell lines Hep2c and on HeLa, human colon tumor cell lines SW1116, HT29, Colon-26, HCT-116 and Caco-2, on the human gastric carcinoma cell line KatoIII, human muscle carcinoma cell lines RD and on C2C12, the human leukemia cell line U937 and B-lymphocyte *tumor cell* lineRp9, on the human lung cancer cell line A549, human neuroblastoma cell line IMR32 and human estrogen positive breast cancer cell line MCF-7, on the human estrogen negative breast cell line MDA-MB-468, MDA-MB-435 and the human non-cancerous breast cell line MCF-10A, as well as on the murine fibroblast cell line L2OB. Concerning the antitumor mechanism, concentrated juice from fruits of *P. domestica* tested on Caco-2 cells resulted a the reduction in the cell viability and nucleosomal DNA fragmentation too, suggesting an apoptotic mechanism [163]. The present MTS cytotoxicity assay on Caco-2 cells at 24 h of cell exposure also indicated the ability of the 50% ethanolic extracts from the fresh plums (EEP at t = 0 and t = 7) to inhibit the human colon carcinoma cell viability in vitro; inhibitory effects were noticed in the interval from 6.42 to 51.4 µg GAE/mL sample, with an inflection point at 12.50 and 25.70 µg GAE/mL sample, corresponding to the extracts at t = 0 and t = 7. Similar to the extracts from strawberry fruits, EEP induced the decrease in the Caco-2 cell viability in the interval from 6.43 to 12.50 µg GAE/mL sample (at t = 0) and 6.43 to 25.70 µg GAE/mL sample (at t = 7), respectively, thus achieving a 52% and 49% inhibition efficacy; at bigger concentrations, the viability of the Caco-2 cells started to increase, but under the negative control line. The MTS anti-proliferative assay at 48 and 72 h, respectively, indicated that starting with the two inflection points noticed in the cytotoxicity study (6.43 µg/mL sample at t = 0 and 12.85 µg GAE/mL sample at t = 7), an increase in the Caco-2 cell viability has occurred; the stimulatory potency at the maximum concentration level in the study (51 µg GAE/mL sample, mean value, *n* = 6) was computed at up to 690% magnitude in the case of the plum extract at t = 0, and up to 390% in the case of the plum extract at t = 7. This suggests that ethanolic extracts from fresh plums, but most probably the fresh plum juice too, are major stimulants of colon (cancer) cells in humans.

The products from grapes, mainly the grape pomace (GP) beverages, are some of the most studied fruits in relation with human health consumption and benefits [164,165,166,167,168,169,170,171,172,173,174,175,176,177,178,179]. Regarding the organ targeted effects, the literature data lean towards major benefits in counteracting the cardiac and coronary heart diseases [164,165,166,167,168,169,170,171,172,173]. As is well-known, the low-density lipoprotein (LDL) cholesterol oxidation in the blood vessels is the basis of the atherosclerosis process, further responsible for the inflammatory response from the arterial vessel wall in humans; thus, studies on ischemic heart disease in rats with induced atherosclerosis through an atherogenic diet indicated that the red grape pomace has the ability to decrease the TNF-α and IL-10 levels, to increase the level of the anti-atherogenic high-density lipoprotein (HDL) cholesterol and also to reduce the size and number of atherosclerotic lesions in rats [167]. The grape pomace and grape polyphenols each reduced the level of ROS, oxidized glutathione (GSSG), thiobarbituric acid (TBARS) and superoxide anion (O^2-^) radicals, and at the same time boosted the activity of the superoxide dismutase (SOD) and catalase (CAT) enzymes, and glutathione peroxidase (GPx) and glutaredoxins (GRx) levels, revealed as the main factors in the prophylaxis of coronary diseases [169]. Resveratrol and its derivatives found in grapes have been proven to attenuate ischemia and reperfusion injury in neonatal cardiomyocytes and protect cardiac tissue in experimental malignant hypertension by decreasing mitochondrial reactive oxygen species generation in human cardiomyocytes [170,171,172,173,174,175]; resveratrol increases the lifespan of eukaryotic cells (up to 60%) [176], but at high concentration levels in medium there were observed harmful effects as well [177]. Regarding antitumor effects, recent anti-proliferative and cytotoxicity screening studies [178] on the crude white pomace/cW, crude red pomace/cR and crude grape seed/cG tested on two colorectal cancer cell lines, Caco-2 and HT29, and one fibroblast line CRL2072 (as control) have shown the anti-proliferative potency of grape-derived products upon Caco-2 cells; cW was the most potent test extract in the study, achieving up to 27.35% cell viability inhibition at the minimal effective concentration in the study (75 μg/mL sample). By the lactate dehydrogenase (LDH) method, cG was found as the most effective extract in the study, especially at the higher concentration (250 μg/mL). The emergence of swollen cells and cytoplasmic vacuolization, together with the loss of the nuclear content and cytoplasmic membrane integrity disruption, were all noticed in the study; therefore, this is likely the mechanism of the cytotoxic activity. Studies have also revealed that the damage upon HT-29 was less pronounced, while the purified extracts appeared to have a higher activity than the crude extracts; finally, the purified grape extracts showed a higher specificity on the colorectal cancer cells compared to the fibroblast cell line CRL2072 [178]. Other studies on several phenolic-rich extracts from biotransformed grape pomace tested on colorectal cancer cell lines Caco-2 and SW-620 have also proved the ability of the GP products at 1.75 mg/mL and 2.5 mg/mL to inhibit the viability of the tested cells by more than 60%; the effects upon the Caco-2 line were less pronounced than those upon the SW620 line [179]. The present MTS cytotoxicity assay after 24 h of cell exposure also indicated the ability of the 50% ethanolic extracts from the fresh green seedless table grapes (EEG at t = 0 and t = 7) to inhibit the growth of the human colon carcinoma cell line Caco-2 along the entire interval tested: from 3.59 to 28.68 µg GAE/mL sample, respectively. The maximum inhibitory potency (computed at up to 79% and 83% at t = 0 and t = 7, respectively) was measured in the interval from 21.51 to 28.68 µg GAE/mL sample. Yet, similar to the other four series of extracts in the study, the MTS anti-proliferative assays at 48 and 72 h indicated that up to 28.68 µg GAE/mL sample, EEG at t = 0 and t = 7 both induced the stimulation of Caco-2 viability in vitro; at the maximum concentration level in the study (29 µg GAE/mL sample, mean value, *n* = 6), a stimulatory effect up to 34% was measured, in comparison with both the positive and negative control series in the present study.

## 4. Materials and Methods

### 4.1. Fruits’ Derived Products Preparation

The present studies were carried out on five series of fresh fruits from a national retail network in Romania, purchased in June 2023 on the day of their sale; the five fruits are strawberries (Romania), cherries (Romania), apricots (Turkey), plums (South Africa) and Thomson seedless green table grapes (Czech Republic). Each series of fresh fruits was washed with water from the city’s network, after which the fruits were placed on absorbent paper towels. The whipped fruits were placed into cardboard boxes in batches close to 100 g, after which they were weighted at two decimals’ precision and divided into the two experiments (*n* = 3): the batches for the experimentation at the initiation time (t = 0) and the batches for the experimentation at the end of the shelf life, t = 5 (strawberries and cherries) and t = 7 (apricots, plums, green table grapes), respectively. The batches at t = 5/7 were placed in the refrigerator in the fruit/vegetable area, after which they were daily inspected to observe their general appearance and to measure the dynamic of the dehydration process along the shelf life. The batches at t = 0 as well as at t = 5/7 were identically processed to obtain the two series of active products in the study: the total acetone extractable and the total outcome fruit waste. Briefly, each batch in each test series was precisely weighted, after which they were mortared in a ceramic/glass mortar cooled at 4 °C; the resulted mortared fruit batches were each extracted four times consecutively with 200 mL of acetone solvent so as to obtain the maximum discoloration of the test fruit and at the same time the minimum coloration of the acetone solvent. The resulting cumulative filtered acetone extracts obtained from each fruit batch in the series (100 g) were each concentrated in a Buchi Rotavapor to a *spiss* type product; the *spiss* product was passed into ethanol to obtain 100 mL 50% ethanolic extract (EE) from each test batch. Thus, at the end, five series of triplicates of 50% ethanolic extracts at t = 0 and five series of 50% ethanolic extracts at t = 5/7 resulted; the extracts were named EES1-3/strawberries, EEC1-3/cherries, EEA1-3/apricots, EEP1-3/plums and EEG1-3/green seedless table grapes. The resulting outcome fruit waste after the repetitive acetone extraction of each fruit batch was dried in a glass desiccator under vacuum, at room temperature (16–18 °C); the resulting products (five series at t = 0, and five series at t = 5/7), namely the acetone powders (AP), in triplicate, were named APS1-3/strawberries, APC1-3/cherries, APA1-3/apricots, APP1-3/plums and APG1-3/green seedless grapes. Thus, the five series of EE/AP products, in triplicate samples, at t = 0 and t = 5/7, respectively, were used for further analytical and pharmacological studies. Appendix A presents the technological aspects along the separation of the two test products from the five fruits’ series in the study; the acetone extract and the resulting fruit waste, namely acetone powder from strawberries, cherries, apricots, plums and green seedless table grapes.

### 4.2. Fruit-Derived Products’ (EE and AP) Analytical Characterization

The EE products were analyzed concerning the total phenolic content and dynamics along the shelf life; the AP products were analyzed concerning the content in minerals and microelements and the elements distribution in waste by comparison with the literature data. The total phenolic content was assessed by the Folin–Ciocalteau method, as described in the European Pharmacopoeia [180] using a UV/Vis Hélios γ (Thermo Electron Corporation, Waltham, MA, USA) spectrophotometer. The content of the total phenolic acids in samples was estimated by comparison with the gallic acid (reference substance) calibration curve: R^2^ = 0.986, *n* = 3. The results are therefore expressed as mg gallic acid equivalents [GAE] per 100 g fresh fruit (mg%, FW). The contents of minerals and microelements in the AP products were assessed by the ICP-MS method (PerkinElmer^®^ ELAN DRC-e ICP-MS, Waltham, MA, USA); the microwave digestion of the test samples was carried out using a model Multiwave™ 3000 microwave system (Anton Paar, Graz, Austria); the precise digestion and analytical conditions in the study are available in the authors’ studies [181]. The chemicals (sodium carbonate), reagents (*Folin-Ciocalteau*), solvents (ethanol, acetone) and the reference compound (gallic acid, 95% purity) used in the study were purchased from the Sigma-Aldrich (Saint Louis, MO, USA) distributor in Romania.

### 4.3. Antioxidant Activity and ROS Efficacy of EE Products

The reactive oxygen species scavenger properties of the EE extracts were appraised by the chemiluminescence (CL) method, luminol–H_2_O_2_ system, at pH = 8.6 [66,67]. The CL tests were performed on the series samples at t = 0 and t = 5/7 to estimate the antioxidant activity changes along the shelf life of the tested fruits; the tests were achieved in comparison with the negative control series using 50% ethanol solvent for triplicate samples (*n* = 3). Briefly, each 50 μL aliquot of 50% ethanolic extract in the EE series, at t = 0 and t = 5/7, respectively, was mixed with 200 μL of 10^−5^ M luminol (prepared in DMSO), 700 μL 0.2M Tris-HCl pH 8.6 (prepared in bi-distilled water) and 50 μL 10^−3^ M H_2_O_2_ (fresh reagent, prepared in bi-distilled water); the negative control series was prepared by using 50% ethanol solvent. The CL reaction initiation, performed by adding H_2_O_2_ in the medium, means that the more antioxidant in a sample, the more strongly and the greater the degree to which it neutralizes the free radicals generated by the oxygenated water decomposition; Luminol reacts to the free radicals remained in the environment by emitting fluorescence; the fluorescence emitted, measured as arbitrary units (a.u.) at the initiation time (3–5 s) and during 60 s total time of reaction, achieve a minimum plateau when the reaction is consumed. The results, a.u. along the time and a.u. along the dilution series, indicate the effectiveness of the reactive scavenger oxygen activity of test samples by comparison with the values obtained for the negative control sample. Thus, the antioxidant activity (AA%) of each test sample can be computed by using the mean value of the a.u. measured at exactly 5 s after the reaction initiation in comparison with the negative control sample. The IC_50_ value of each test sample is quantified by the investigation of a.u. along the dilution series so as to achieve the inflection moment in which the maximum antioxidant activity at the minimum concentration of the test product is obtained (8 to 12 serial dilutions are usually necessary). This way, the effectiveness on scavenging hydroxyl radicals in the medium of the five series of fruits extracts at t = 0 and t = 5/7, respectively, were compared and also the magnitude of the AA% and IC_50_ values, to achieve the data regarding their changes along the shelf life.

### 4.4. In Vitro Pharmacological Studies on the EE Products

In vitro studies were performed by MTS assay following the CellTiter 96AQueous One Solution Cell Proliferation Assay Promega Corporation (Madison, WI, USA) protocol [68]; the MTS assay is based on the selective ability of viable cells in culture to reduce 3-(4,5-dimethylthiazol-2-yl)-5-(3-carboxymethoxyphenyl)-2-(4-sulfophenyl)-2H-tetrazolium (MTS) in the medium into purple-colored formazan crystals which can be further measured at 492 nm. The MTS assay can be performed as a cytotoxicity or anti-proliferative activity experiment. To achieve the status of a cytotoxicity activity experiment, the cells are exposed to the test and control samples at the time when a “semiconfluent” cell culture is achieved (around 70% cell proliferation), while in the anti-proliferative activity experiment the application of the test and control samples is performed at the time when a “sub-confluent” cell culture is achieved (around 30% cell proliferation). Briefly, the test extracts are prepared as a dilution series in the interval designed to be relevant for the active compounds in the study, with each point as a triplicate sample; in the present study, it was designed the interval between 1 and 55 µg GAE/mL sample, which means 5 to 50 µL EE extract per sample. In parallel, the same dilution series was prepared for the solvent in which the test sample was prepared, thus obtaining the positive control series (namely 50% ethanol); the negative control series usually contains the cell culture medium in the study. This way, after reaching the particular cell confluence in the study (70% for cytotoxicity and 30% for anti-proliferative), the cells are detached from the flask with Trypsin–EDTA. The cell suspension is further centrifuged at 2000 rpm for 5 min, after that the cells resulted are re-suspended in the Caco-2 growth medium (20% heat-inactivated FBS, Merck, Sigma-Aldrich, Saint Louis, MO, USA, distributor in Romania); the cells were then seeded in 96-well plates at a density of 8000 and 4000 cells per well, respectively, in 200 μL culture medium. After 20, 44 h or 68 h of the cell exposure to the test/control samples, the culture medium is removed. The cells were then incubated with MTS solution for another two hours and, after that, the viability of the adherent cells is estimated by exactly measuring the absorbance of the solutions at 492 nm (BOECO BMR-100 Microplate Reader, Hamburg, GER). The optical densities (O.D. at 492 nm) measured (mean value, *n* = 3) can be used as such or as cell viability percentages along the dilution series, by reporting the test series to positive/negative control series. The results were computed for their statistical significance (Student “*t*” test). In the present study: the notation (*) means results without statistical significance and *p* > 0.05; the notation (**) means results with statistical significance and 0.05 < *p* < 0.01; the notation (***) means results with statistical significance and *p* < 0.01.

## 5. Conclusions

Overall, studies on the five perishable fruits (strawberries, cherries, apricots, plums and green seedless table grapes) at the initiation time (t = 0) and at the end of the experiment (t = 5/7 days) indicated only small changes consisting of either an increase or decrease in the chemical and biological attributes studied: polyphenols dynamic, antioxidant efficacy and the effects upon human colon cancer cells (Caco-2) in vitro, respectively. From the consumer’s side, this means that the five perishable fruits can be safely processed in home-made derived products beyond the 5–7 days of shelf life, if they are kept in the refrigerator at a 4 °C temperature.

According to the specialized literature data, the jelly (gelatin) form provides the best bioavailability of the secondary metabolites in humans.

Studies on the antioxidant activity dynamic indicated a decrease in the ROS (particularly hydroxyl radical) scavenging efficacy of apricots (−11%), plums (−6%) and strawberries (−4%), while cherries and green table grapes gained 7% and 2% antioxidant potency at the end of the shelf life, respectively.

The cytotoxicity/anti-proliferative MTS assays on Caco-2 cells exposed to ethanolic extracts from the five fruits in the study indicated inhibitory but also stimulatory potency of the fruits’ series upon the human colon carcinoma cell viability in vitro.

These results confirm the antioxidant and the antitumor power of the edible fruits in the literature data, while the most probable explanation of the stimulatory potency consists of the fact that once the fuel compounds (sugars) and the micronutrients exceed a certain concentration in the environment, the cancer cells likely use the fruits’ extracts to their advantage. Accordingly, it is expected that the separated fractions, polyphenol-rich extracts deprived of sugars, no longer present the stimulatory potency.

## Figures and Tables

**Figure 1 ijms-25-04848-f001:**
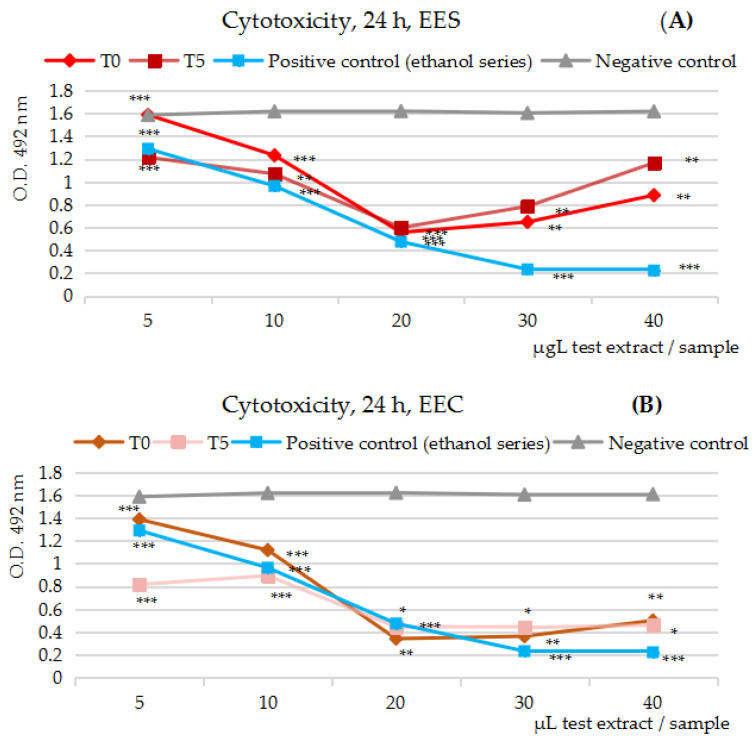
(**A**–**E**) The effects of the 50% ethanolic extracts from the five fruits’ series at t = 0 and t = 5/7 tested on the human colon tumor cell line Caco-2 in the cytotoxicity MTS assay at 24 h (h) of exposure; the O.D. variation at 492 nm along the dilution series by comparison with the negative control series (gray line) and positive control series (blue line). (**A**) The effects of 50% ethanolic extracts from strawberries/EES upon Caco-2 cell viability. (**B**) The effects of 50% ethanolic extracts from cherries/EEC upon Caco-2 cell viability. (**C**) The effects of 50% ethanolic extracts from apricots/EEA upon Caco-2 cell viability. (**D**) The effects of 50% ethanolic extracts from plums/EEP upon Caco-2 cell viability. (**E**) The effects of 50% ethanolic extracts from green seedless table grapes/EEG upon Caco-2 cell viability. Notation: * = results without statistical significance (*p* > 0.05); Notation: ** = results with statistical significance (0.01 < *p* < 0.05, *n* = 3, mean values); Notation: *** = results with statistical significance (*p* < 0.01, *n* = 3, mean values).

**Figure 2 ijms-25-04848-f002:**
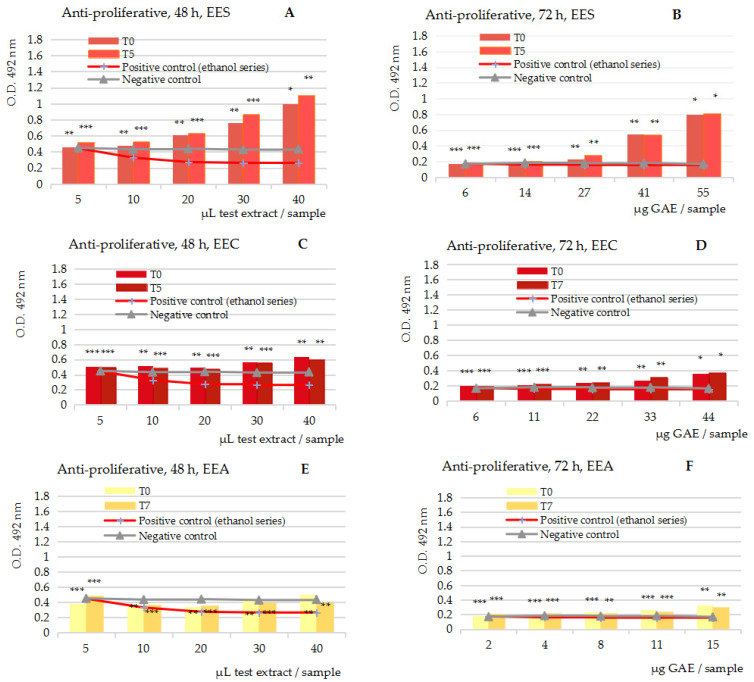
(**A**–**J**) The effects of the 50% ethanolic extracts from the five fruits’ series at t = 0 and t = 5/7 tested on the human colon tumor cell line Caco-2 in an anti-proliferative MTS assay at 48 h and 72 h of exposure; the O.D. variation at 492 nm along the dilution series (expressed as µL test extract/sample at 48 h and µg GAE/sample at 72 h) by comparison with the negative control series (gray line) and positive control series (red line). (**A**,**B**) The effects of 50% ethanolic extracts from strawberries/EES upon Caco-2 cell viability. (**C**,**D**) The effects of 50% ethanolic extracts from cherries/EEC upon Caco-2 cell viability. (**E**,**F**) The effects of 50% ethanolic extracts from apricots/EEA upon Caco-2 cell viability. (**G**,**H**) The effects of 50% ethanolic extracts from plums/EEP upon Caco-2 cell viability. (**I**,**J**) The effects of 50% ethanolic extracts from green seedless table grapes/EEG upon Caco-2 cell viability. Notation: * = results without statistical significance (*p* > 0.05); Notation: ** = results with statistical significance (0.01 < *p* < 0.05, *n* = 3, mean values); Notation: *** = results with statistical significance (*p* < 0.01, *n* =3, mean values).

**Table 1 ijms-25-04848-t001:** Major compounds reported in the five test fruits.

Strawberry (*Fragaria ananassa*)	Sweet Cherry (*Prunus avium)*	Apricot (*Prunus armenica*)	Plum (*Prunus domestica)*	Green Grape (*Vitis vinifera)*
pelargonidin-3-*O*-glucoside cyanidin-3-*O*-glucoside pelargonidin-3-*O*-rutinoside catechin epicatechin chlorogenic acid caffeic acid cinnamic acid 3,4,5-methoxycinnamic acid protocatechuic acid p-coumaric acid ellagic acid salicylic acid vanillic acid ferulic acid iso-ferulic acid pyrogallol catechol coumarin [31,32,33,34,35]	cyanidin-3-*O*-rutinoside cyanidin-3-*O*-glucoside peonidin-3-*O*-glucoside peonidin-3-*O*-rutinoside pelargonidin-3-*O*-rutinoside catechin epicatechin epigallocatechin epicatechin gallate epigallocatechin gallate hydroxycinnamic acid quercetin apigenin luteolin chrysin eriodyctiol hesperitin naringenin genistein daidzein glycitein formononectin [36,37,38,39]	caffeic acid chlorogenic acid neochlorogenic acid cyanidin-3-*O*-glucoside procyanidins B1, B2, B3 catechin epicatechin p-coumaric acid gallic acid ferulic acids kaempferol myricetin quercetin-3-*O*-rutinoside lutein zeaxanthin violaxanthin β-cryptoxanthin α-, β-, γ-carotene [40,41,42,43,44,45]	chlorogenic acid caffeic acid cyanidin peonidin catechin epicatechin quercetin derivatives [46,47]	procyanidins B1,B2,B3,B4, procyanidins C1 procyanidins EEC catechin epicatechin epigallocatechin gallocatechin epicatechin-3-*O*-gallate isorhamnetin-3-*O*-glucoside kaempferol-3-*O*–galactoside kaempferol-3-*O*–glucoside quercetin-3-*O*-glucuronide quercetin-3-*O*-rutinoside caffeoyl tartaric acid cis-caffeoyl tartaric acid cis-*p*-coumaroyl tartaric acid p-coumaroyl tartaric acid trans-caffeoyl tartaric acid trans-p-coumaroyl tartaric acid resveratrol, resveratrol-3-*O*-glucoside, trans-resveratrol [48,49]

**Table 2 ijms-25-04848-t002:** The total phenolics content in 50% ethanolic extracts from the five fruits’ series (EE), expressed as mg GAE per 100 g FW, at the beginning (t = 0) and at the end of the experiment (t = 5/7).

Test Batches along the Shelf Life (days)	EES	EEC	EEA	EEP	EEG
	mg GAE/100 g Fresh Fruit (Mean Values, mg%, FW)				
t = 0	129 ± 3	118 ± 4	33 ± 1	123 ± 2	74 ± 2
t = 5/7	134 ± 4	110 ± 3	37 ± 1	131 ± 3	72 ± 2
Total phenolics’ dynamic (%)	+4.40%	−7.74%	+11.04%	+6.15%	−1.72%
Dehydration process’ dynamic (%)	−3.0%	−10.1%	−11.70%	−2.7%	−1.3%

*n* = 3; ± SD; *p* < 0.01.

**Table 3 ijms-25-04848-t003:** The content of the main minerals and microelements in the acetone powders (AP products) from the five fruits’ series, expressed as mg per 100 g fresh fruit (mg%, FW).

Minerals and Microelements Content in Batches	APS	APC	APA	PP	APG
	mg/μg * per 100 g Fresh Fruit (Mean Values, mg%, FW)				
K	41 ± 1.5	59 ± 2.1	59 ± 2.3	34 ± 2.5	62 ± 2.8
Ca	11 ± 0.52	9 ± 0.69	7.5 ± 0.43	4 ± 0.68	5.3 ± 0.61
Mg	6 ± 0.24	5.5 ± 0.41	4 ± 0.35	2.6 ± 0.32	2.3 ± 0.38
Fe	0.41 ± 0.03	0.24 ± 0.01	0.13 ± 0.01	0.16 ± 0.01	0.16 ± 0.01
Cu *	15 ± 0.15	44 ± 3.08	32 ± 2.24	32 ± 1.89	18 ± 1.26
Zn *	66 ± 4.62	92 ± 6.44	77 ± 5.39	63 ± 4.41	84 ± 5.88
As *	3.3 ± 0.10	5.28 ± 0.16	2.87 ± 0.09	2.32 ± 0.07	2.28 ± 0.07
Pb *	4.75 ± 0.14	5.76 ± 0.17	5.49 ± 0.16	3.29 ± 0.10	3.60 ± 0.11
Total elements in AP series	58	74	71	40	70

*n* = 3, ± SD, *p* < 0.05, * Cu, Zn, As and Pb are expressed as µg%, FW.

**Table 4 ijms-25-04848-t004:** The distribution of the minerals and microelements in the five test fruits; a comparison of the literature data [50] and the data on the acetone powders (AP products) in the present study.

Test Element	Location	Fresh Fruits Content vs. Vegetal Waste Content in the Study (AP); mg/100 g Fresh Fruit and the Percent (%) in the Vegetal Waste				
		Strawberries	Cherries	Apricots	Plums	Green Grapes
K	fresh fruit/literature data waste/acetone powder (AP)	183 41 (22%)	248 59 (24%)	283 59 (21%)	197 34 (17%)	203 62 (30%)
Ca	fresh fruit/literature data waste/acetone powder (AP)	15 11 (73%)	20 9 (45%)	20 7.5 (38%)	8 4 (50%)	14 5.3 (38%)
Mg	fresh fruit/literature data waste/acetone powder (AP)	13 6 (46%)	10 5.5 (55%)	12 4 (33%)	7 2.6 (37%)	8 2.3 (29%)
Fe	fresh fruit/literature data waste/acetone powder (AP)	0.5 0.41 (82%)	0.5 0.24 (48%)	0.8 0.13 (16%)	0.3 0.16 (53%)	0.3 0.16 (53%)
The percent of test elements in the AP series		APS-28%	APC-27%	APA-22%	APP-19%	APG-31%

**Table 5 ijms-25-04848-t005:** A comparison of AA% and IC_50_ of the five series of 50% ethanolic extracts at t = 0 and t = 5/7.

Test Extract	AA (%)		IC_50_ (μg GAE/μL Sample)		IC_50_ Variation (%)
	T = 0	T = 5/7	T = 0	T = 5/7	
EES	99	99	3.22	3.35	−4%
EEC	99	99	5.93	5.50	+7%
EEA	99	99	1.67	1.86	−11%
EEP	99	99	3.08	3.27	−6%
EEG	99	99	3.70	3.64	+2%

## Data Availability

Data are contained within the article and Appendix A.

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
