# Peer review of "A View on the Chemical and Biological Attributes of Five Edible Fruits after Finishing Their Shelf Life: Studies on Caco-2 Cells"

_ijms, 2024, doi:10.3390/ijms25094848_

Round 1

Reviewer 1 Report

Comments and Suggestions for Authors

The manuscript under appreciation deals with a topic of interest, that is the exploitation of fruit waste to evaluate its antioxidant potential and helath benefits. The authors for this purpose used tumor Ca-co2  cells in vitro and studied the extracts of different fruits. 

The concept and the hypothesis of the study are novel. In fact, recent trends in the literature adopt this hypothesis for the evaluation of any health benefits using fruits or foods of plant origin even thermally processed. The data given are convincing and have been discussed. However, there are some technical drawbacks that must be corrected through a revision. More specifically, the Introduction and Conclusion sections are too extended and need re-organization. Some recent literature that have already been published supporting this idea must be cited. In addition, the authors must provide superscripts in the tables (where is required) to show the statistically significant differences among the different fruits studies. Overall, the paper is good and can be accepted after a revision. For this purpose, I have enclosed my suggestions within the attached pdf.

Finally, figures and tables are ok (a graphical abstract of the study could be also provided).

Comments on the Quality of English Language

The English language needs moderate revision

Author Response

The authors thank you for very careful and professional guidance, through which the article gained a significant improvement. Thus: 

  • The Introduction and Conclusion sections are too extended and need re-organization. - REVISED
  • Some recent literature that have already been published supporting this idea must be cited. - REVISED in Introduction part (References 10 - 23)
  • The authors must provide superscripts in the tables (where is required) to show the statistically significant differences among the different fruits studies. - REVISED (Table 2 and 3)
  • A graphical abstract of the study could be also provided (attached).

Reviewer 2 Report

Comments and Suggestions for Authors
Review IJMS-2964258

Though the work contains some useful information, the manuscript should go through significant revision/rewritten to enhance the readability and clarity. Many sections like introduction and discussion should be more concise, with more mechanistic references to support.

Title: must be improved. The current version is too much detailed.

The dietary flavonoids on health role is mainly via regulating gut microbiota. Thus, the discussion should not be limited to the total amount only. The authors can search database like Web of Science with ‘dietary flavonoids health role’ to get reference for further discussion.

There were detailed purification steps for some ingredient analyses. Thus, what’ the possible influence of the matrix effect? Matric compounds removal is vital for the assay accuracy and reliability. The authors need to search database titled with ‘matrix compounds removal assaying’ to get reference for further discussion.    

Fruit derived products, antioxidants, and ROS etc should be linked to each other. They are changing within the fruit as metabolism is going on. Some further discussion on food metabolomics can be performed for quality analyses. The authors can search database like Web of Science titled with ‘food metabolomics quality analyses’ to get reference to improve the discussion. 

Author Response

The authors thank you for very careful and professional guidance, through which the article gained a significant improvement. Thus:

  • The manuscript should go through significant revision/rewritten to enhance the readability and clarity.- REVISED
  • Many sections like introduction and discussion should be more concise, with more mechanistic references to support. - REVISED
  • Title: must be improved. The current version is too much detailed. - REVISED
  • The dietary flavonoids on health role is mainly via regulating gut microbiota… The authors can search database like Web of Science with ‘dietary flavonoids health role’ to get reference for further discussion. - REVISED in Discussion part (references 62 - 64)
  • What’ the possible influence of the matrix effect? - REVISED in analytical part (references 14 - 21)
  • Some further discussion on food metabolomics can be performed for quality analyses. ..The authors can search database like Web of Science titled with ‘food metabolomics quality analyses’ to get reference to improve the discussion. - REVISED in Introduction part (references 22 - 23).

Round 2

Reviewer 2 Report

Comments and Suggestions for Authors
Review IJMS-2964258-v2 

The authors have addressed most of the questions quite well. The quality of the revised manuscript significantly. However, there are still two questions not addressed well. 

Firstly, the authors applied References 62-64 to address the question on dietary flavonoids. However, they are not much related to dietary flavonoids and the mechanism behind-regulating gut microbiota. The discussion should not be limited to the total amount only. The authors can search database like Web of Science titled with ‘dietary flavonoids health role’ to get more related reference to enhance the discussion.​

Secondly, the explanation and discussion on matrix effect need further revision. The references applied mainly mentioned that the testing methods were effective, achieved good matrix removal effect. However, the science behind and logical procedures were not much addressed. It would be good to discuss further with specific reference focusing on robust procedures including blank matrix sample, matrix-matched standards, matrix-matched standard calibration curves and matrix effect. The authors are suggested to review the question again and search database like Web of Science titled ‘matrix compounds removal for assaying’ to get more appropriate reference to improve the discussion.

Author Response

The article has been revised in accordance with the two questions addressed and we are confident that it now meets the conditions to be published.

Round 3

Reviewer 2 Report

Comments and Suggestions for Authors

Review ijms-2964258-v2

The authors have addressed the questions quite well. The quality of the revised manuscript has been improved significantly. There are no further comments. The current version is acceptable for publication.